**Data Availability Statement:** All relevant data of this study will be made available upon study completion.

# What are the impacts of oral complications from cancer therapy on the quality of life of children? A protocol to update a scoping review

Nona Attaran[1‡], Apoorva Sharma[1☉], Martin Morris[2☉], Olawale Dudubo[1☉], Mary Ellen Macdonald[3]*

1 Faculty of Dental Medicine and Oral Health Sciences, McGill University, Montreal, Québec, Canada, 2 Schulich Library of Physical Sciences, Life Sciences, and Engineering, McGill University, Montreal, Québec, Canada, 3 Faculty of Medicine, Dalhousie University, Halifax, Nova Scotia, Canada

☉ These authors contributed equally to this work.
‡ NA share first author on this work.
* maryellenmacdonald@dal.ca

## Abstract

### Introduction

Cancer treatments can damage healthy tissues and organs, and leave harmful impacts on cancer survivors, especially on children and adolescents. The oral effects of cancer treatment can occur during or soon after treatment, or months–even years–later. Cancer treatments can also affect the child, psychologically and socially by hindering their speech, eating, sleeping, and social interactions. These effects can have profound impacts on children's quality of life. Building on a previous review published in 2012, this scoping review aims to identify and map the current evidence base underpinning the oral health-related impacts of cancer treatment on the quality of life of children with cancer.

### Methodology and methods

Our methodology is guided by Arksey and O'Malley's methodological framework for scoping reviews, Levac's additions to the framework, and follows the Joanna Briggs Institute Reviewer's Manual. Five electronic databases and grey literature will be systematically searched using a predefined search strategy. Two reviewers will independently screen the retrieved articles using Rayyan software and chart data from included articles. One of the team's senior research members will act as a third reviewer and make the final decision on disputed documents. We will include literature with a focus on oral health-related quality of life of children undergoing cancer treatments. Following the selection of studies, data will be extracted, synthesized, and reported thematically and the relevant stakeholder's insight will be added to our results.

**Funding:** The author(s) received no specific funding for this work.

## Introduction

According to the World Health Organization (WHO), each year approximately 400,000 children and young adults are diagnosed with cancer globally [1]. The most common types of childhood cancer are leukemia, lymphomas, brain cancer, and solid tumors, including neuroblastoma and Wilms tumor [2]. Whereas childhood cancer mortality used to be dire, today upwards of 80% can be cured with treatments such as pharmaceuticals, surgery, and radiotherapy [1]. Notwithstanding the vast improvement in mortality rates, cancer treatments can leave devastating impacts on cancer survivors, damaging healthy tissues and causing systemic side effects. These effects can be especially devastating for young people who experience high rates of radiotherapy and chemotherapy-induced complications [3]. A common location for side effects is the oral cavity, including the soft and hard tissues in the mouth, from the lips anteriorly to faucial pillars posteriorly [4, 5].

Treatment-related oral side effects can occur during or soon after treatment, or months–even years–later. They are classified as early (acute) effects and late (chronic) effects [6]. Early oral effects include oral mucositis, xerostomia (dry mouth), oral infections (e.g., candidiasis and herpes virus infections), and taste disturbances. Late effects include dental decay, and abnormalities in dental and jaw development. Combined treatments such as combination of chemotherapy with radiation increase the risk of dental problems such as dental caries, taste disturbances, and missing teeth or roots [5]. These conditions influence the physical, functional, and psychosocial health of children who have undergone cancer therapy, and can continue to affect them months or years after the treatment is completed [3].

Evidence has highlighted the profound impact that oral health can have on a child's quality of life [7–10]. Specific to childhood cancer, a scoping review by Noronha and Macdonald on the oral effects of cancer treatment demonstrated the devastating impact of such effects on the quality of life of children [9]. Children in the reviewed studies experienced both early and late oral effects as follows:

- Mucositis was the most common early effect in this review, affecting almost 100% of children undergoing chemotherapy. (8) In addition to pain, mucositis has psychological and social impacts; for example, speech issues associated with oral mucositis can reduce a child's engagement in social interaction [11].

- Xerostomia is a result of damage to the salivary gland which changes the consistency and amount of saliva in the mouth. It can be an early side-effect when caused by chemotherapy and can have a long-term effect when caused by radiotherapy to the head and neck. Decreased salivary flow and increased viscosity can cause difficulty with chewing, swallowing, speech, and also affect the function of taste buds resulting in taste alteration, causing a dislike for some foods and appetite loss. This taste alteration can result in nausea, vomiting, pain, and discomfort [5].

- After mucositis and xerostomia, taste disturbance after chemotherapy were found to be the next most common effect [12]. These children were more sensitive to bitterness, and had taste recognition errors, which affected their appetite. This appetite change can lead to malnutrition and impair quality of life by affecting appetite, body weight, and psychological well-being [13].

- According to this review, more than 80% of children treated for cancer had at least one longer-term dental anomaly (e.g., root shortening, smaller teeth, enlarged pulp chambers). These malformations arise during remission and can hinder eating, speech, and social interactions, and can require additional complex clinical interventions [9].

Besides the valuable knowledge that was gathered through Noronha and Macdonald's review there were limitations stated by the original review, as well as limitations that have become apparent in the decade since that review was published.

Since this scoping review, the oral health-specific measure, oral health-related quality of life (OHRQoL), has become more commonly used in studies on the effects of oral diseases and oral complications on patients' oral symptoms. This multidimensional construct focuses on how an individual's oral health affects their comfort, abilities and well-being (e.g., eating, sleeping, social interactions, self-esteem) [14]. The increased use of OHRQoL follows the growing recognition of oral health as an essential component of systemic health and general wellbeing [15, 16]. As oral health is strongly age-dependent, and therefore OHRQoL in children is different from adults, this measure has been adapted for child populations [17]. We therefore aim to assess whether (and how) there has been attention to this construct in recent years in relation to the oral health of children surviving cancer.

Further, according to this prior review, there was a dearth of qualitative research into the experiences of children about how oral side effects of cancer therapy impacts their quality of life [12]. This result is not surprising; in a 2007 review of pediatric oncology studies, 85% of studies on childhood cancer did not solicit patient-reported outcomes, instead relying on parent and health care professional's reports [18]. While having the parent's and health care providers' perspective is clearly important, it has become evident that children's perspectives are not always consistent with adults. As a result, a new approach to child-focused research has started to engage children directly in research to better understand their experiences first-hand [19]. This movement is consistent with article 12 of The United Nations Convention on the Rights of the Child which stipulates that children's experiences must be rendered through their own voices and that they have a right to express their own views in matters that affect them [20]. While dental research has started to follow this trend [21], it is not known if or how research on the impact of cancer treatment on children's quality of life has followed suit since Noronha and Macdonald's review up to 2011. Moreover, we are especially interested to see if and how research approaches can acknowledge and incorporate the unique insights, experiences, and perspectives of children surviving cancer on their OHRQoL.

Further by updating the 2011 review, we seek to investigate if and how there has been growth in literature on treatments regarding both cancer, and oral health sequalae. We have also expanded the scope of our study to include adult survivors of childhood cancer. By doing so, we aim to include the long-term impact of cancer treatment complications on Quality of Life and oral health; this was not incorporated in the original review despite knowledge of long-term effects.

Further in this review we intend to incorporate a stakeholder consultation (e.g., health care professionals, children with cancer, parents), as part of this literature review in comparison to the original review. By adding the stakeholder's insight and perspective to the evidence gained from the review, strategies for knowledge mobilization and uptake can be developed. Following is the procedure we will follow to conduct this scoping review.

Therefore, the aim of this scoping review is to scope the literature since 2011 for the impacts on quality of life due to therapy-related oral complications on childhood cancer survivors and the children's contribution in producing this knowledge.

## Materials and methods

A scoping review is an exploratory research method that scopes the literature on a given topic and identifies gaps in the current research and highlight areas that require further inquiry [22]. The purpose of conducting a scoping review is to identify the types of available evidence in a

given field, clarify key concepts in the literature, examine how research is conducted on a certain topic or field, and identify and analyze knowledge gaps [22].

This protocol will follow the Joanna Briggs Institute Reviewer's Manual to assure transparency, accuracy, and completeness [23]. We will also follow Arksey and O'Malley's methodological framework, which consists of six stages for conducting a scoping review: 1. Identifying the research question; 2. Identifying relevant studies; 3. Selecting studies; 4. Charting the data; 5. Collating, summarizing, and reporting of results; and 6. Consulting with relevant stakeholders [24] (Fig 1).

All members of the research team developed, reviewed, and agreed with this protocol; we intend to complete the review by Fall 2023.

## 1. Identifying the research question

Building on the work of Noronha and Macdonald mentioned above, the main objective of this review is to map and synthesize the knowledge on the impact of oral complications from

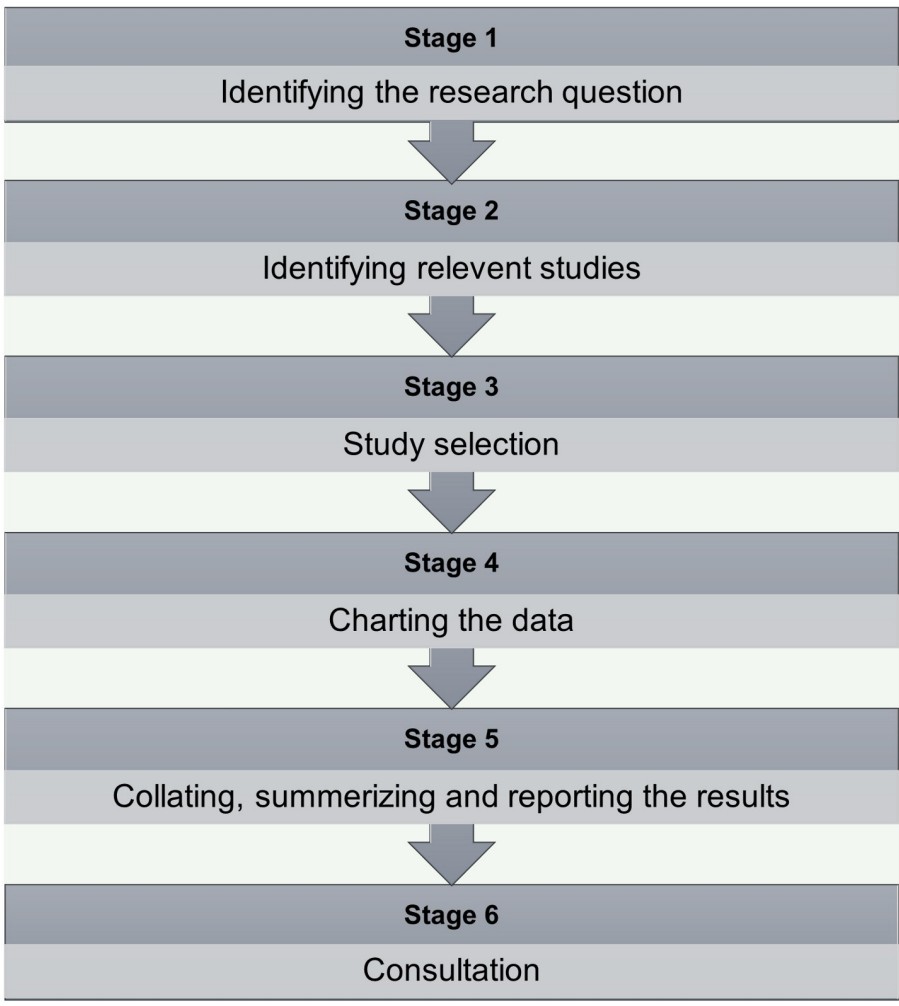

**Fig 1. Arksey and O'Malley's methodological framework.**

cancer therapy on the quality of life of children surviving cancer starting in 2011. Our primary research questions is:

What are the impacts of oral complications from cancer therapy on the quality of life of childhood cancer survivors?

And our secondary research questions will be: How are children involved in producing knowledge related to the effects of cancer treatment on their oral health related quality of life?

## 2. Identifying relevant studies

The identification of relevant literature will consist of several combined approaches, including searching electronic database and grey literature sources, and reference list screening. Articles will be accessed through five electronic databases: MEDLINE/PubMed, Scopus, Embase, Web of Science, and PsychInfo. A librarian (MM) has created the search strategy; (Table 1) he will lead citation management and assist with search documentation. The initial search strategy has been piloted to verify breadth, comprehensiveness, and feasibility. This search strategy will be adapted and applied to each database. We will review the reference lists of included studies to identify relevant studies that were not identified in the initial search.

**Table 1. Search strategy.**

| Search Strategy |
|---|
| 1. exp Antineoplastic Agents/ |
| 2. exp Radiotherapy/ |
| 3. exp Hematopoietic Stem Cell Transplantation/ |
| 4. exp Bone Marrow Transplantation/ |
| 5. (antineoplastic or chemotherap* or radiotherap* or ((h?ematopoietic or bone marrow) adj3 (SCT or transplant*))).tw,kw. |
| 6. or/1-5 |
| 7. Oral Health/ or exp Dentistry/ or Halitosis/ or exp Stomatognathic Diseases/ or DMF Index/ or Periodontal Index/ |
| 8. (dentist* or endodont* or orthodonti* or periodont* or prosthodont* or apicoectom* or gingivectom* or gingivoplast* or glossectom* or "mandibular advancement" or alveolectom* or alveoloplast* or vestibuloplast* or "root canal" or (oral adj1 (care or health or hygiene or surgical or surgery or mucositis)) or oropharyng* or temporomandibular or TMJ or jaw or jaws or mandibular or maxillofacial or mandible* or maxilla* or "alveolar ridge" or dental or orthognathic or tooth or teeth or occlusion or malocclusion or mal-occlusion or odontolog* or tongue* or glossal or buccal or palatal or palate or palates or labial or lip or lips or gingiva* or gingiviti* or saliva* or DMF).tw,kw. |
| 9. 7 or 8 |
| 10. "Quality of Life"/ or exp rehabilitation/ or exp eating/ or exp human activities/ or (rh or px).fs. |
| 11. (quality of life or well-being or long-term or (daily adj1 (life or living)) or rehabilitat* or depress* or pain or immunosuppress* or "disease management" or "Child Oral Health Impact Profile" or C-DAS or CFSS-DS or COHRQoL or COHIP or CPQ or ECOHIS or FIS or OASIS or OHQoL or OHRQoL or QOL or P-CPQ or POQL or (("Early Childhood Oral Health Impact" or "Oral Aesthetic Subjective Impact" or "Corah Dental Anxiety" or "Family Impact") adj1 Scale) or "Children's Fear Survey Schedule" or (("Child" or "Parental-Caregiver") adj1 "Perceptions Questionnaire")).tw,kw. |
| 12. 10 or 11 |
| 13. 6 and 9 and 12 |
| 14. limit 13 to "all child (0 to 18 years)" |
| 15. exp Child/ or exp Pediatrics/ |
| 16. (infan* or toddler* or minors or boy? or boyhood or girl? or child* or schoolchild* or school child* or adolescen* or juvenil* or youth* or teen* or under*age* or p?ediatric*).tw,kf. |
| 17. 15 or 16 |
| 18. 13 and 17 |
| 19. 14 or 18 |
| 20. limit 19 to (english or french) |

## 3. Selecting studies

In this stage, all retained studies will be merged into a single Endnote library with duplicated articles removed. The merged Endnote library will be imported into Rayyan software (Qatar Computing Research Institute, QCRI) for screening. The study screening and selection process will be conducted by two members of the research team (AS and NA). The two reviewers will be calibrated: they will independently assess titles and abstracts of the first 50 studies according to the inclusion and exclusion criteria, after which inter-rater reliability will be measured using Cohen's k coefficient. The calculated coefficient will act as an indicator of whether reviewers understand and apply the inclusion criteria consistently. If there is low agreement (<0.40), the reviewers will consult, and, if needed, adjust or reword the eligibility criteria. This process will be repeated until inter-rater agreement reaches substantial levels (>0.40).

Screening and selecting studies will then consist of 2 phases during which the reviewers will assess study inclusion against a set of predefined eligibility criteria outlined in Table 2.

The first study selection phase includes the title and abstract screening of all identified documents. A third reviewer (OD) will assist in the selection process if the two primary reviewers cannot reach a consensus. Titles and abstracts that appear to meet the eligibility criteria will be retained. The second study selection phase consists of full-text review of studies that have been classified as potentially eligible during phase one. Each reviewer will review the full text of the selected articles and put their comments about the reason of including or excluding the study in Rayyan to be seen by other reviewers. Any disagreements concerning the eligibility of the articles will be solved through discussion between the reviewers with the appointment of a third reviewer if required. This stage will include an iterative process, incorporating searching of the literature, refinement of search strategies, and selection of articles.

## 4. Charting the data

Arksey and O'Malley's methodological framework suggests charting the data according to central research themes. Thus, we will develop a data extraction tool in line with the review's' objectives and corresponding research questions. We anticipate the data extracted will include author, publication year, location of study, study design, age of children, type of cancer, type of treatment, type of oral complications, OHRQoL measurement tools and assessment strategies,

**Table 2. Inclusion and exclusion criteria.**

| | **Inclusion criteria** |
|---|---|
| Population | • Data specific to children, 0-18yrs<br>• Studies with adults who had cancer during childhood (0-18yrs) |
| Intervention | • All types of cancer treatment interventions<br>• Any type of childhood cancer<br>• Data specific to oral health complications from cancer treatment |
| Outcome | • The outcome consists of any information related to quality of life and oral health related quality of life. |
| Study type | • Any primary study type<br>• Publications between 2011 and 2021.<br>• English or French publications. |
| | **Exclusion criteria** |
| | • Studies involving a mixed sample of adults and children<br>• Studies in which the disease category was not well defined or defined as mixed diseases (e.g., studies that the child has both a cancer and a non-cancer disorder) |

the type of children's involvement in the study, and findings related to the effect of cancer treatment on the oral health-related quality of life of the children.

To ensure that all relevant data are extracted, the tool used for data extraction will be reviewed by the two reviewers prior to implementation. Differences in the suggested information included will be discussed between reviewers (if necessary, with a third reviewer) in a meeting to reach agreement. Furthermore, to ensure the tool's utility, consistency with the research questions and purpose and, the agreement level between reviewers, it will be piloted on 10 articles by both reviewers and any needed modifications will be implemented.

In the charting phase, reviewers will compare their extracted data. Inconsistencies and disagreements will be discussed, reconsulting the respective documents and if necessary, requesting support by a senior researcher of the team. Further, the tool will be iteratively updated if necessary, during the study's full extraction process, with any modifications detailed in the full scoping review report. Finally, throughout the process, there will be weekly team meetings during which ambiguities, concerns or other issues will be discussed.

## 5. Collating, summarizing, and reporting findings

The primary goal of a scoping review is to present a comprehensive summary of current evidence and significant findings across various domains [25]. Therefore, the chosen analytical approach will be descriptive and narrative, aligning with the study objectives. We will follow the three steps outlined by Levac et al. [25] for this stage:

First, we will report the data using a descriptive numerical summary and thematic analysis to describe the characteristics of the included studies. The PRISMA (Preferred Reporting Items for Systematic reviews and Meta-Analyses) extension for scoping reviews (PRISMA-ScR) statement outlines a minimum set of items to include in scoping review reports to increase methodological transparency and uptake of research findings [26]. According to the PRISMA-ScR statement, a description of the study selection process will be presented in a flow diagram format (Fig 2) [23]. This diagram will aid replicability and transparency. The extent, scope and nature of retained literature will be summarized descriptively using ranges and counts, presented in graphs, charts, or tables according to our charting categories. This step will provide an overview of existing evidence and research activity trends, as well as highlight potential research gaps [24].

Second, we will report the results specifically regarding the impact of cancer therapy on the OHRQoL of children and their involvement in producing this knowledge in studies. We will use graphs, charts, or tables where useful and provide an accompanying narrative summary to highlight how the results are linked to the objectives and research questions of this study.

Finally, we will identify the knowledge gaps and the broader implications for future research, policy and practice.

## 6. Consultation with relevant stakeholders

While stakeholder consultation is considered optional in Arksey and O'Malley's framework [24] we believe it will provide additional valuable insights into our findings and opportunities for knowledge transfer in the field of pediatric oncology and pediatric dentistry. Thus, we will proceed by convening health care professionals (e.g., oral health professionals; oncological professionals) working for children surviving cancer, parents of these children and preferably the children themselves. By consulting and sharing the scoping study findings with these stakeholders and adding the experience of these groups we can gain insights that can lead to improved study outcomes.

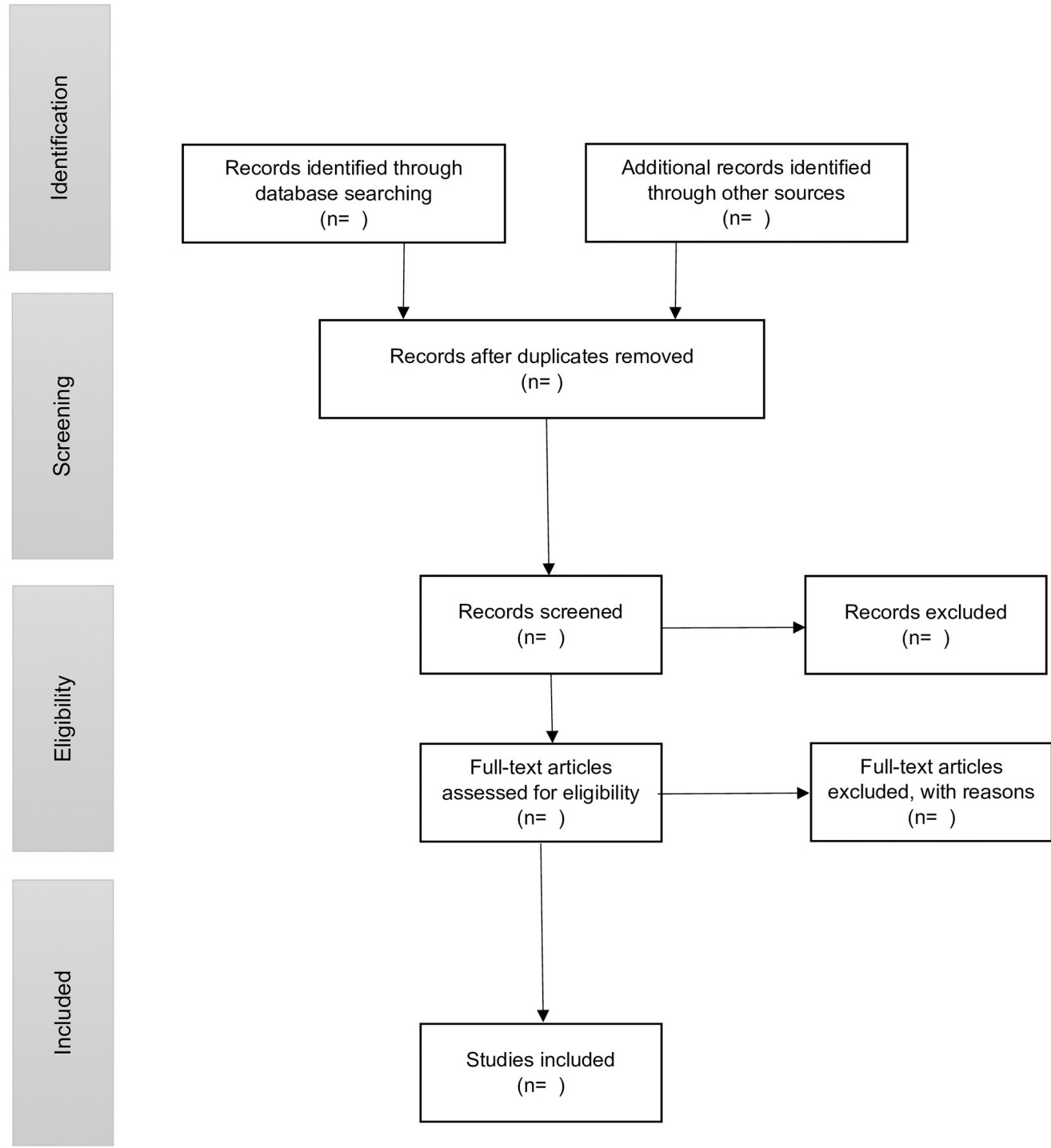

**Fig 2. PRISMA-ScR flow diagram.**

**Ethics.** For the initial review, ethical approval will not be required as there are no human participants involved. We will seek ethical approval specifically for the Stakeholder consultation; details will be determined after preliminary analysis is completed.

## Discussion

In this scoping review, we aim to explore the impact of oral complications resulting from cancer treatment on the quality of life of children who are going through or have survived cancer. Our goal is to gain a better understanding of how these complications affect their well-being and quality of life. Thus this review will update a previous literature review, providing comprehensive information on the impacts of cancer treatment on the oral health related quality of life of children undergoing cancer treatments by identifying, synthesizing, and summarizing the reported literature in the past ten years [9]. Our preliminary review suggests that in the past ten years, the literature has grown substantially regarding children's OHRQoL. Moreover, as quality of life is a subjective concept, we hypothesize we will find more children's direct involvement in reporting consequences related to their OHRQoL. We also anticipate our review will show areas that have been under-researched and may require further investigation and evaluation.

While the primary focus of treatment remains on combating cancer, mitigating the side effects of treatment can contribute to enhancing the quality of life of pediatric cancer survivors. Recognizing the influence of cancer treatment-related oral complications on quality of life can assist healthcare professionals in providing more holistic care and addressing the challenges faced by both patients and their families.

The results of this review will be shared through cancer and oral health conferences and symposia to disseminate the knowledge. Additionally, the findings will be published and shared with relevant stakeholders. The information extracted from this review will serve as a foundation for a qualitative study, focusing on the impacts of oral health effects of childhood cancer on quality of life, with a specific emphasis on gathering perspectives from children themselves.

## Supporting information

**S1 Checklist. PRISMA-P 2015 checklist.**
(PDF)

## Author Contributions

**Conceptualization:** Nona Attaran, Apoorva Sharma, Mary Ellen Macdonald.

**Data curation:** Nona Attaran, Apoorva Sharma, Martin Morris, Olawale Dudubo, Mary Ellen Macdonald.

**Formal analysis:** Nona Attaran, Apoorva Sharma, Mary Ellen Macdonald.

**Funding acquisition:** Nona Attaran, Apoorva Sharma, Mary Ellen Macdonald.

**Investigation:** Nona Attaran, Apoorva Sharma, Martin Morris, Olawale Dudubo, Mary Ellen Macdonald.

**Methodology:** Nona Attaran, Apoorva Sharma, Martin Morris, Mary Ellen Macdonald.

**Project administration:** Nona Attaran, Apoorva Sharma, Martin Morris, Mary Ellen Macdonald.

**Resources:** Nona Attaran, Apoorva Sharma, Mary Ellen Macdonald.

**Software:** Nona Attaran, Apoorva Sharma, Mary Ellen Macdonald.

**Supervision:** Martin Morris, Mary Ellen Macdonald.

**Validation:** Nona Attaran, Apoorva Sharma, Martin Morris, Olawale Dudubo, Mary Ellen Macdonald.

**Visualization:** Nona Attaran, Apoorva Sharma, Martin Morris, Olawale Dudubo, Mary Ellen Macdonald.

**Writing – original draft:** Nona Attaran, Apoorva Sharma, Mary Ellen Macdonald.

**Writing – review & editing:** Nona Attaran, Apoorva Sharma, Martin Morris, Olawale Dudubo, Mary Ellen Macdonald.

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
