## [Decision Letter · Decision Letter 0]

4 Sep 2023

PONE-D-23-25040What are the impacts of oral complications from cancer therapy on the quality of life of children? A protocol to update a scoping review.PLOS ONE

Dear Dr. attaran,

Thank you for submitting your manuscript to PLOS ONE. After careful consideration, we feel that it has merit but does not fully meet PLOS ONE’s publication criteria as it currently stands. Therefore, we invite you to submit a revised version of the manuscript that addresses the points raised during the review process.

We look forward to receiving your revised manuscript.

Kind regards,

Yolanda Malele-Kolisa, BDS, MPH, MDent, PhD

Academic Editor

PLOS ONE

Journal Requirements:

Additional Editor Comments:

The protocol is well received. It needs to meet certain criteria for it to be accepted for publication. See Reviewers comments.

Reviewers' comments:

Reviewer's Responses to Questions

**Comments to the Author**

1. Does the manuscript provide a valid rationale for the proposed study, with clearly identified and justified research questions?

Reviewer #1: Yes

Reviewer #2: Partly

2. Is the protocol technically sound and planned in a manner that will lead to a meaningful outcome and allow testing the stated hypotheses?

Reviewer #1: Yes

Reviewer #2: Yes

3. Is the methodology feasible and described in sufficient detail to allow the work to be replicable?

Reviewer #1: Yes

Reviewer #2: Yes

4. Have the authors described where all data underlying the findings will be made available when the study is complete?

Reviewer #1: Yes

Reviewer #2: Yes

5. Is the manuscript presented in an intelligible fashion and written in standard English?

Reviewer #1: Yes

Reviewer #2: Yes

6. Review Comments to the Author

You may also provide optional suggestions and comments to authors that they might find helpful in planning their study.

Reviewer #1: Your proposed methods for updating a scoping review are also well explained and feasible, and the potential contribution of your research is well articulated.

Reviewer #2: The author needs to address certain questios and provide a bit more detail on the following;

Why the need to update the 2011 review?

What makes your work different ? What gap are you closing?

What was done to address the quality of life of the children after the findings of 2011??? Why are there reviews on reviews if no impactful solution has not been brought about??

7. PLOS authors have the option to publish the peer review history of their article (what does this mean?). If published, this will include your full peer review and any attached files.

Reviewer #1: No

Reviewer #2: **Yes: **Mpho Matlakale Molete

---

## [Author Response · Author response to Decision Letter 0]

19 Sep 2023

Response Letter

September 10, 2023

Yolanda Malele-Kolisa, 

Academic Editor

PLOS ONE

Dear Academic Editor

Subject: Submission of revised paper (PONE-D-23-25040)

Thank you for your email dated 4th September 2023 enclosing the reviewers’ comments. We have carefully reviewed the comments and have revised the manuscript accordingly. Our responses are given in a point-by-point manner below. Changes to the manuscript are shown in red.

We have addressed the editorial points as follows:

1. As the corresponding author, we reviewed PLOS ONE's style requirements and ensured that our manuscript met them thoroughly. We Changed the file names according to your guidelines. Resubmitted the figures and supportive files according to your standards and went through the manuscript file to make necessary changes according to your requirements. 

2. As for the Data Availability Statement according to your response to our email dated 12th Sep we changed the Data Availability Statement in the revised manuscript attached. (Case Number:08181908 ref:00DU0Ifis._500PM1S7HT:ref)

3. As the corresponding author, I ensured that my ORCID iD is validated in Editorial Manager.

We hope the revised version is now suitable for publication and look forward to hearing from you in due course.

Sincerely,

Nona Attaran

Corresponding author 

Response to Reviewer 1:

Thank you for your review of our paper. We truly appreciate the time you have put into our manuscript!

Response to Reviewer 2:

Thank you for your review of our paper. We truly appreciate the time you have put into our manuscript and answered each of your points below. We also added some explanation to the revised manuscript to make it more clear in rational.

Why the need to update the 2011 review? What makes your work different? What gap are you closing? 

The need to update the 2011 review stemmed from several key objectives; these objectives were built from the limitations stated by the original review, as well as limitations that have become apparent in the decade since that review was published. They are as follows: 

• The original review did not specifically search for the term ‘Oral Health-Related Quality of Life.’ This term was new a decade ago; it has gained prominence in recent years. We therefore aim to assess whether (and how) there has been attention to this construct.

• Most studies in the original review relied on parents and caregivers providing information on behalf of the children. Given that Childhood Studies has grown in attention and urgency, we aim to assess if (and how) there has been a shift towards including children themselves directly in research. By directly incorporating their perspectives and experiences, we believe clinicians and policymakers will be better placed to understand the repercussions of cancer treatment on children’s oral health and consequently their overall QoL.

• The original literature review reported few qualitative studies. Recognizing QoL as ultimately a subjective concept, we are especially interested to see if and how research approaches can acknowledge and incorporate the unique insights, experiences, and perspectives of children surviving cancer on QoL.

• We have also expanded the scope of our study to include adult survivors of childhood cancer. By doing so, we aim to include the long-term impact of cancer treatment complications on QoL and oral health; this was not incorporated in the original review despite knowledge of long-term effects. 

• New treatments have been developed since the original review (e.g., regarding both cancer, and oral health sequalae). By updating the 2011 review, we seek to investigate if and how there has been growth in literature on treatments.

What was done to address the quality of life of the children after the findings of 2011? 

• The original review was not well mobilized in terms of knowledge translation; we know of no specific dissemination interventions designed after that review and therefore do not believe that it had a measurable impact on the QOL of children. Our intention, through incorporating a stakeholder consultation, is to both gain insight through consulting with stakeholders (e.g.,  healthcare professionals, children with cancer, and parents), as well as to work with them to develop strategies for knowledge mobilization and uptake.   

Why are there reviews on reviews if no impactful solution has not been brought about? 

• Some problems are inherently complex without a single, straightforward solution. In such cases, ongoing discussions and further reviews can help uncover nuances and adapt strategies over time. Further, the circumstances surrounding a problem can change, requiring a reevaluation of previous solutions. Finally, unique stakeholders, such as children, parents, healthcare professionals, and policymakers, may each have different perspectives on what constitutes an impactful solution. Reviews on reviews can help consolidate these perspectives and find common ground. 

• In essence, while it may seem frustrating that no impactful solution has yet been achieved, reviews on reviews can be part of a larger process of problem-solving. They can contribute to eventual change by helping refine research and dissemination approaches, and by fostering ongoing discussions and collaborations amongst stakeholders.

---

## [Decision Letter · Decision Letter 1]

11 Oct 2023

What are the impacts of oral complications from cancer therapy on the quality of life of children? A protocol to update a scoping review.

PONE-D-23-25040R1

Dear Dr. Attaran,

We’re pleased to inform you that your manuscript has been judged scientifically suitable for publication and will be formally accepted for publication once it meets all outstanding technical requirements.

Kind regards,

Yolanda Malele-Kolisa, BDS, MPH, MDent, PhD

Academic Editor

PLOS ONE

Additional Editor Comments (optional):

Reviewers' comments:

Reviewer's Responses to Questions

**Comments to the Author**

1. Does the manuscript provide a valid rationale for the proposed study, with clearly identified and justified research questions?

Reviewer #2: Yes

2. Is the protocol technically sound and planned in a manner that will lead to a meaningful outcome and allow testing the stated hypotheses?

Reviewer #2: Yes

3. Is the methodology feasible and described in sufficient detail to allow the work to be replicable?

Reviewer #2: Yes

4. Have the authors described where all data underlying the findings will be made available when the study is complete?

Reviewer #2: Yes

5. Is the manuscript presented in an intelligible fashion and written in standard English?

Reviewer #2: Yes

6. Review Comments to the Author

You may also provide optional suggestions and comments to authors that they might find helpful in planning their study.

Reviewer #2: Thank you for the clarification regarding the justification of your study. The protocol reads much better now.

7. PLOS authors have the option to publish the peer review history of their article (what does this mean?). If published, this will include your full peer review and any attached files.

Reviewer #2: **Yes: **Mpho Matlakale Molete

---

## [Editor Report · Acceptance letter]

8 Nov 2023

PONE-D-23-25040R1 

What are the impacts of oral complications from cancer therapy on the quality of life of children? A protocol to update a scoping review. 

Dear Dr. MacDonald:

I'm pleased to inform you that your manuscript has been deemed suitable for publication in PLOS ONE. Congratulations! Your manuscript is now with our production department. 

Kind regards, 

on behalf of

Prof Yolanda Malele-Kolisa 

Academic Editor

PLOS ONE